# Bioassay-Guided Isolation of New Flavonoid Glycosides from *Platanus* × *acerifolia* Leaves and Their *Staphylococcus aureus* Inhibitory Effects

**DOI:** 10.3390/molecules27175357

**Published:** 2022-08-23

**Authors:** Xiying Wu, Yu Tang, Ezzat E. A. Osman, Jiang Wan, Wei Jiang, Guoxun Yang, Juan Xiong, Quangang Zhu, Jin-Feng Hu

**Affiliations:** 1Department of Natural Medicine, School of Pharmacy, Fudan University, Shanghai 201203, China; 2Institute of Natural Medicine and Health Products, School of Pharmaceutical Sciences, Zhejiang Provincial Key Laboratory of Plant Ecology and Conservation, Taizhou University, Taizhou 318000, China; 3Shanghai Skin Disease Hospital, Tongji University School of Medicine, Shanghai 200443, China; 4College of Pharmaceutical Sciences, Zhejiang University, Hangzhou 310058, China; 5Department of Medicinal Chemistry, Theodor Bilharz Research Institute, Kornaish El-Nile St., Giza 12411, Egypt; 6School of Life Science and Technology, Wuhan Polytechnic University, Wuhan 430023, China

**Keywords:** *Platanus* × *acerifolia*, Platanaceae, flavonoid glycosides, antibacterial, *Staphylococcus aureus*

## Abstract

Despite the rapid advances in drug R&D, there is still a huge need for antibacterial medications, specifically for the methicillin-resistant *Staphylococcus aureus* (MRSA). Inspired by the research where a viable class of MRSA inhibitors was found in the species *Platanus occidentalis*, a *S. aureus* inhibition screening-guided phytochemical reinvestigation on *Platanus* × *acerifolia* (London plane tree) leaves were performed with four flavonoid glycosides garnered, including two new compounds, quercetin-3-*O*-α-l-(2″-*E*-*p*-coumaroyl-3″-*Z*-*p*-coumaroyl)-rhamnopyranoside (*E*,*Z*-3′-hydroxyplatanoside, **1**) and quercetin-3-*O*-α-*l*-(2″-*Z*-*p*-coumaroyl-3″-*E*-*p*-coumaroyl)-rhamnopyranoside (*Z*,*E*-3′-hydroxyplatanoside, **2**). All of the isolates showed significant *S. aureus* ATCC 25904 inhibitory activity with MICs ranging from 4 to 64 μg/mL, suggesting the potential of discovering drug leads for the control of *S. aureus* from such a rich, urban landscaping plant in the *Platanus* genus.

## 1. Introduction

Owing to the long-standing misuse of antibiotics, *Staphylococcus aureus* has become resistant to many commonly used first-line drugs [1,2]. For instance, methicillin-resistant *S. aureus* (MRSA) is able to resist most of the *β*-lactam antibiotics and is a leading cause of high bacterial infection-related morbidity and mortality [3,4,5]. There is consequently an urgent need to develop alternatives for the control of *S. aureus* infections. Still, plant-derived natural products remain one of the primary sources of new drugs [6,7], as demonstrated by the isolation and characterization of a few flavonoid glycosides with promising anti-MRSA bioactivity from the American sycamore, *Platanus occidentalis* (family Platanaceae), as presented by Hamann et al. [4]. The genus *Platanus* contains seven deciduous species and twelve subspecies [8], of which three are widely distributed in China, namely *P. occidentalis* Linn., *P. orientalis* Linn. (the oriental plane) and *P.* × *acerifolia* Willd (the hybrid London plane) [9,10]. Of note, *P. occidentalis* and *P. orientalis* have long been used as traditional folk medicines for the treatment of a variety of diseases such as blepharitis, conjunctivitis, gastrointestinal disorders, diarrhea, toothache and skin disease [4,11]. Being one of the most famous urban landscaping trees in the world, *P.* × *acerifolia* is popular, and which earned the reputation of the “King of Road Trees” [12,13].

Reportedly, a wide range of secondary metabolites (e.g., flavonoids, glycosides, furan coumarins, terpenoids, sterols and organic acids) have been isolated from the buds [8,14,15,16,17,18,19,20] and the bark [21,22,23,24,25] of *P. occidentalis* and *P. orientalis*. While being very much abundant in China, only a few flavonols have been obtained from the leaves of *P.* × *acerifolia* [26]. Literally, from both chemodiverse and antibacterial perspectives, *P.* × *acerifolia* leaves can be considered underexplored. In addition, as inspired by the research regarding the discovery of a viable class of MRSA inhibitors mentioned above [4], we therefore conducted a phytochemical study on *P.* × *acerifolia* leaves under the guidance of a *S. aureus* inhibitory activity screening.

## 2. Results and Discussion

The preliminary bioactivity evaluation of subfractions (F1–F6) from the 75% ethanolic extract of *P.* × *acerifolia* leaves revealed that F4 (60% EtOH) showed a mild inhibitory effect (MIC: 256 μg/mL) on *S. aureus* ATCC 25904. This subfraction was then subjected to repeated column chromatography over silica gel, macroporous resin D101, MCI gel and Sephadex LH-20 followed by semi-preparative HPLC to afford two new (**1** and **2**) and two known (**3** and **4**) flavonoid glycosides as shown in Figure 1. The new ones were elucidated to be quercetin-3*-O*-α-(2″-*E-p*-coumaroyl-3″-*Z-p*-coumaroyl)- rhamnopyranoside (**1**) and quercetin-3*-O*-α-(2″-*Z-p*-coumaroyl-3″-*E-p*-coumaroyl)-rhamnopyranoside (**2**) through the 1D-/2D-NMR data analysis in conjunction with the HRMS experiments. By comparing the observed and reported spectroscopic data and physicochemical properties, the known structures were identified as kaempferol-3*-O*-α-l-(2″-*Z-p*-coumaroyl-3″-*E-p*-coumaroyl)-rhamnopyranoside (*Z*, *E*-platanoside, **3**) [4] and kaempferol-3*-O*-α-l-(2″,3″-di-*E-p*-coumaroyl)-rhamnopyranoside (*E*, *E*-platanoside, **4**) [4]. The HRMS and NMR spectra of compounds **1** and **2** are available in Supporting Information.

Compound **1** was obtained as a yellowish-brown powder, and its positive-mode HRESIMS data showed a sodium adduct ion at *m/z* 763.1621 [M + Na]^+^, which established a molecular formula of C_39_H_32_O_15_ with the aid of the ^13^C NMR (Table 1) data. The UV spectrum showed characteristic absorption bands at 272 and 314 nm, suggesting the existence of a 3-OH substituted flavanol skeleton [27,28]. The ^1^H NMR spectrum (Table 1) displayed three coupled aromatic proton signals at *δ*_H_ 7.42 (1H, dd, *J* = 2.0, 8.4 Hz, H-6′), *δ*_H_ 7.38 (1H, d, *J* = 2.0 Hz, H-2′), and *δ*_H_ 6.96 (1H, d, *J* = 8.4 Hz, H-5′) attributed to a 1,3,4-trisubstituted benzene ring, along with two *meta*-coupled doublets at *δ*_H_ 6.39 (1H, d, *J* = 2.0, H-8) and *δ*_H_ 6.21 (1H, d, *J* = 2.0, H-6). Undoubtedly, the data above delineated a quercetin backbone [29,30,31]. In addition, the ^1^H-^1^H COSY motif (Figure 2) of H-1″/H-2″/H-3″/H-4″/H-5″/H_3_-6″ in conjunction with the ^13^C (Table 1) and HSQC NMR experiments (Appendix A) revealed the presence of an α-rhamnose moiety [29,32], which was found to locate at C-3 due to the key HMBC correlation from H-1″ to C-3 (δ*_C_* 135.56) (Figure 2). Besides, two *p*-coumarate units were constructed owing to the observation of two sets of well-resolved proton signals: δ_H_ 7.47 (2H, br d, *J* = 8.4 Hz, H-2‴, 6‴), δ_H_ 6.80 (2H, br d, *J* = 8.4 Hz, H-3‴, 5‴), δ_H_ 7.61 (1H, d, *J* = 15.6 Hz, H-7‴), and δ_H_ 6.31 (1H, d, J = 15.6 Hz, H-8‴) accounting for a (E)-p-coumarate unit; δ_H_ 7.69 (2H, br d, J = 8.4 Hz, H-2′′′′, 6′′′′) and δ_H_ 6.77 (2H, br d, J = 8.4 Hz, H-3′′′′, 5′′′′), together with δ_H_ 6.88 (1H, d, J = 12.8 Hz, H-7′′′′) and δ_H_ 5.73 (1H, d, J = 12.8 Hz, H-8′′′′), arising from a (*Z*)-*p*-coumaryl moiety. The down-field shifted the chemical shifts of H-2″ and H-3″ as well as the HMBC correlations (Figure 2) from H-2″ to C-9‴ (δ_C_ 167.77) and H-3″ to C-9′′′′ (δ_C_ 167.58) allowed the assembly of the (*E*)- and (*Z*)-*p*-coumaryl units that connected to C-2″ and C-3″ via the ester bonds, respectively. Thus, compound **1** was determined as quercetin-3-*O*-α-(2″-*E-p*-coumaroyl-3″-*Z-p*- coumaroyl)-rhamnopyranside.

Compound **2** exhibited the same molecular formula (C_39_H_32_O_15_) as **1** based on the HRESIMS data (Appendix A). A comparison of the NMR data of **2** (Table 2) with those of 1 showed the very much close structural similarity between the two compounds. Differing from **1**, the bonding positions of the (*E*)- and (*Z*)-*p*-coumaryl units with the rhamnose were found to be swapped upon closer inspection of the HMBC spectrum of **2** (Figure 2), with the (*E*)-*p*-coumaryl unit connecting to C-3″, while (*Z*)-*p*-coumaryl unit connecting to C-2″. Thus, compound **2** was identified as quercetin-3*-O*-α-(2″-*Z-p*-coumaroyl-3″-*E-p*-coumaroyl)-rhamnopyranoside.

The ability of compounds **1**–**4** to inhibit the growth of *S. aureus* ATCC 25904 was then tested by in vitro antibacterial susceptibility assays. Methicillin and chloramphenicol were used as the positive controls. Notably, all the compounds were found to possess antibacterial activity against *S. aureus* with MICs ranging from 4 to 64 μg/mL (Table 3). Among them, compounds **3** and **4** showed considerable *S. aureus* inhibition activities with MICs at the level of 4 and 16 μg/mL, respectively. Along with the relatively weaker inhibition effects of **1** and **2**, both MIC values are 64 μg/mL—this study is in good agreement with the reported data in terms of the structure–activity relationship [4]. Briefly, all the compounds in this study showed antibacterial activity, confirming that a flavonoid moiety connected to the *p*-coumaroyl groups via a sugar unit (e.g., rhamnose) and hydroxy groups at positions 5, 7, and 4′ are essential for the antibacterial activity against *S. aureus*. Meanwhile, the bioactivity results of compounds **1** and **2** implied that the presence of a hydroxy group at position 3′ presumably has a negative impact on such antimicrobial activity while the *E-* or *Z-*configuration of the *p*-coumaroyl units does not exert effects. In general, the data presented herein suggest that the genus *Platanus* (especially the abundant *P.* × *acerifolia* leaves) could serve as a promising source for the development of drugs to treat *S. aureus* infections, more specifically, glycosides from this genus with flavonoids and *p*-coumarinoids being the aglycone parts.

## 3. Materials and Methods

### 3.1. General Experimental Procedures and Agents

UV and IR spectra were recorded on a U-2900E spectrophotometer (Hitachi High-Technologies, Beijing, China) and a Nicolet Is5 FTIR spectrometer (Thermo Fisher Scientific, Waltham, MA, USA), respectively. 1D- and 2D-NMR experiments (^1^H, ^13^C, DEPT, COSY, HSQC and HMBC) were performed in CD_3_OD on a Bruker Avance III 400, and/or a Bruker Avance 600 MHz spectrometer (Bruker BioSpin, Rheinstetten, Germany). HRESIMS were acquired on a micro TOF-QII or an AB 5600+ Q TOF spectrometer (Bruker Daltonics, Bremen, Germany). ESIMS were obtained from a Waters UPLC H ClassSQD or an Agilent 1100 series mass spectrometer. CC was performed over silica gel (100–200 mesh, Kang-Bi-Nuo Silysia Chemical Ltd., Yantai, China), MCI (CHP20P, 75–150 Μm, Mitsubishi Chemical Industries, Tokyo, Japan), macroporous resin D101 (Sinopharm Chemical Reagent Co., Ltd., Shanghai, China) and Sephadex LH-20 (GE Healthcare Bio-Sciences AB, Uppsala, Sweden). RP-HPLC separations were conducted on a Waters e2695 system equipped with a Waters 2998 Photodiode Array Detector on ODS columns (SunFire: 5 μm, 150 × 4.6 mm; 5 μm, 250 × 10 mm). TLC analyses were carried out using precoated GF254 (0.25 mm thickness) plates (Kang-Bi-Nuo Silysia Chemical Ltd., Yantai, China); the compounds were detected by UV light (254 and/or 365 nm) and 10% H_2_SO_4_-EtOH.

### 3.2. Plant Material

The green leaves of *P.* × *acerifolia* were collected from dozens of trees in January 2018 along the roadside at Zhangjiang campus, Fudan University in Shanghai, China, and identified by Prof. Ze-Xin Jin (Taizhou University). A voucher specimen (No. 20180508) has been deposited at the Herbarium of the School of Pharmaceutical Sciences, Taizhou University, China.

### 3.3. Extraction and Isolation

*Platanus* × *acerifolia* leaves (30 kg) were dried, ground, and extracted four times using 75% ethanol (EtOH). The 75% ethanolic extract (3.5 kg) was loaded to column chromatography over macroporous resin D101 as stationary phase and eluted with step gradients of EtOH/H_2_O (from 0% to 100%, *v*/*v*) to afford six main fractions: F1 (100% H_2_O), F2 (30% EtOH), F3 (50% EtOH), F4 (60% EtOH), F5 (75% EtOH), and F6 (100% EtOH). F4 (750 g) was re-subjected to D101 column chromatography with step gradient elution of EtOH/H_2_O (from 30% to 100%, *v*/*v*), yielding six sub-fractions F4-A~F. F4-D (70% EtOH, 70.0 g) was then successively purified using MCI column chromatography with MeOH/H_2_O (from 50% to 100%, *v*/*v*) used as the step gradient mobile phase, Sephadex LH-20 CC, and semi-prep RP-HPLC [SunFire; flow rate, 3.0 mL/min]. Finally, compounds **1** (3.0 mg, t*_R_* = 22.5 min) and **2** (3.2 mg, t*_R_* = 31.7 min) were obtained using semi-prep HPLC-DAD [SunFire, MeOH/H_2_O (containing 0.05% TFA, *v*/*v*) 66:34, *v*/*v*], while compounds **3** (5.0 mg, t*_R_* = 19.6 min) and **4** (20.0 mg, t*_R_* = 16.2 min) were generated using semi-prep HPLC-DAD [SunFire, MeOH/H_2_O (containing 0.05% TFA, *v*/*v*) 73:27, *v*/*v*].

Compound **1**: yellowish-brown amorphous powder; UV (MeOH) λ_max_ (log ε) 272 (3.18), 314 (3.46) nm; IR (film) ν_max_ 3419, 2963, 2922, 2853, 1659, 1607, 1512, 1384, 1054 and 1018 cm^−1^; ^1^H and ^13^C NMR (CD_3_OD) data, see Table 1; HRESIMS *m/z* 763.1621 [M + Na]^+^ (calcd. for C_39_H_32_O_15_, 763.1633, δ = −1.6 ppm).

Compound **2**: yellowish-brown amorphous powder; UV (MeOH) λ_max_ (log ε) 270 (3.70), 312 (4.05) nm; IR (film) ν_max_ 3421, 2970, 2923, 2866, 1659, 1605, 1512, 1384, 1050 and 1014 cm^−1^; ^1^H and ^13^C NMR (CD_3_OD) data, see Table 2; HRESIMS *m/z* 763.1613 [M + Na]^+^ (calcd. for C_39_H_32_O_15_, 763.1633, δ = −2.7 ppm).

### 3.4. In Vitro Antibacterial Susceptibility Assays

The minimum inhibitory concentration (MIC) was evaluated on the basis of Clinical Laboratory Standards Institute (CLSI) guidelines by the conventional two-fold microbroth gradient dilution assay [33,34]. *S. aureus* ATCC 25904 was inoculated to the Brain Heart Infusion (BHI) agar plates and cultured for 18~24 h at 35 °C for activation. The diluted bacterial suspension in Cation-adjusted Mueller-Hinton Ⅱ broth (CAMHB) with a turbidity of (1~2) × 10^6^ CFU/mL was ready for detection according to the direct bacterial suspension method. The subfractions and isolates were firstly dissolved in DMSO to produce solutions at the concentrations of 10.0 mg/mL and 2.0 mg/mL, respectively. The resulting solutions were then diluted with fresh CAMHB medium to produce working concentrations of 1024 (from 512 μg/mL to 1 μg/mL in 96-well plates) and 128 μg/mL (from 64 μg/mL to 0.125 μg/mL in 96-well plates), respectively. Then, 100 μL of the working isolates solution was distributed in each well, while the growth controls contained equal amounts of DMSO. Finally, the bacteria-containing suspension (100 μL) was added to each well. The 96-well plates were incubated at 35 °C for 16~20 h, and MIC was determined as the lowest concentration of the drugs that completely inhibited the growth of bacteria. Chloramphenicol and Methicillin were used as positive controls, which were active against *S. aureus*. All the tests were performed in triplicate.

### 3.5. Statistical Analysis

The MIC results were analyzed for their variances by ANOVA using the SAS program (version 9.2). Differences among means of different compounds were assessed by Tukey’s multiple comparison statistical analysis at a *p* = 0.05 level of significance.

## 4. Conclusions

Previous phytochemical studies on the buds [8,14,15,16,17,18,19,20] and the bark [21,22,23,24,25] of *P. occidentalis* and *P. orientalis* led to the isolation and characterization of a large number of secondary metabolites, including flavonoids, glycosides, furan coumarins, terpenoids, sterols and organic acids. However, as a hybrid of these two medicinally used plants [4,11], *P*.× *acerifolia* is still lacking phytochemical and biological investigations to a large extent. In the present work, we hence focused on the flavonoid glycosides from the leaves of *P*.× *acerifolia*, which gleaned two previously undescribed flavonoid glycosides (compounds **1** and **2**) as well as two known ones (compounds **3** and **4**). Regarding the bioactivity evaluations, all of the isolates showed considerable *S. aureus* ATCC 25904 inhibition effects with MICs ranging from 4 to 64 μg/mL. While the mechanism of action requires further study, which is currently in progress in our laboratory and would trigger the next stage of research, including animal testing and the survey of plant resources available for providing bulk drug substances, the above findings expanded the structural diversity of *P*.× *acerifolia* and could provide useful clues for discovery and development of new therapeutic or preventive agents for the treatment of *S. aureus* infection-related diseases.

## Figures and Tables

**Figure 1 molecules-27-05357-f001:**
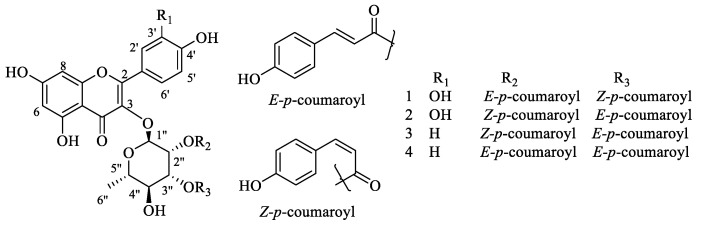
Chemical structures of flavonoid glycosides **1**–**4** from *P.* × *acerifolia* leaves.

**Figure 2 molecules-27-05357-f002:**
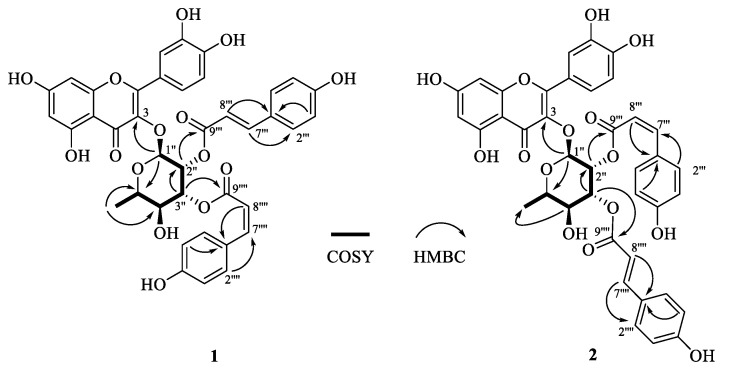
Key 2D (COSY and HMBC) NMR correlations of compounds **1** and **2**.

**Table 1 molecules-27-05357-t001:** NMR spectroscopic data (in CD_3_OD) for compound **1**.

Position	δ_C_	δ_H_, Multi. (*J* in Hz)	Position	δ_C_	δ_H_, Multi. (*J* in Hz)
Flavonol moiety			Rhamnosyl moiety		
2	161.45		1″	100.35	5.53, d (1.6)
3	135.56		2″	70.80	5.81, dd (1.6, 3.6)
4	179.35		3″	72.83	5.25, dd (3.6, 9.6)
5	163.25		4″	70.91	3.58, t (9.6)
6	99.89	6.21, d (2.0)	5″	72.27	3.60, m
7	165.92		6″	17.74	1.03, d (5.2)
8	94.75	6.39, d (2.0)	1‴	127.08	
9	158.55		2‴, 6‴	131.40	7.47, br d (8.4)
10	105.88		3‴, 5‴	116.84	6.80, br d (8.4)
1′	122.76		4‴	159.29	
2′	116.69	7.38, d (2.0)	7‴	147.57	7.61, d (15.6)
3′	146.61		8‴	114.29	6.31, d (15.6)
4′	149.96		9‴	167.77	
5′	116.63	6.96, d (8.4)	1′′′′	127.58	
6′	123.05	7.42, dd (2.0, 8.4)	2′′′′, 6′′′′	133.85	7.69, br d (8.4)
			3′′′′, 5′′′′	115.87	6.77, br d (8.4)
			4′′′′	160.11	
			7′′′′	145.84	6.88, d (12.8)
			8′′′′	116.20	5.73, d (12.8)
			9′′′′	167.58	

**Table 2 molecules-27-05357-t002:** NMR spectroscopic data (in CD_3_OD) for compound **2**.

Position	δ_C_	δ_H_, Multi. (*J* in Hz)	Position	δ_C_	δ_H_, Multi. (*J* in Hz)
Flavonol moiety			Rhamnosyl moiety		
2	161.31		1″	100.49	5.49, d (1.6)
3	135.65		2″	70.62	5.81, dd (1.6, 3.6)
4	179.38		3″	73.11	5.25, dd (3.6, 9.6)
5	163.98		4″	70.91	3.51, t (9.6)
6	99.90	6.22, d (2.0)	5″	72.22	3.61, m
7	165.34		6″	17.69	1.01, d (6.0)
8	94.76	6.40, d (2.0)	1‴	127.37	
9	158.56		2‴, 6‴	133.98	7.63, br d (8.8)
10	105.53		3‴, 5‴	115.99	6.63, br d (8.8)
1′	122.73		4‴	160.22	
2′	116.68	7.39, d (2.0)	7‴	146.58	6.92, d (12.8)
3′	146.61		8‴	115.37	5.79, d (12.8)
4′	149.97		9‴	166.58	
5′	116.61	6.97, d (8.4)	1′′′′	127.10	
6′	123.02	7.43, dd (2.0, 8.4)	2′′′′, 6′′′′	131.24	7.36, br d (8.8)
			3′′′′, 5′′′′	116.83	6.77, br d (8.8)
			4′′′′	160.47	
			7′′′′	147.10	7.63, d (16.0)
			8′′′′	114.94	6.28, d (16.0)
			9′′′′	168.65	

**Table 3 molecules-27-05357-t003:** In vitro antibacterial activity of the isolated flavonoid glycosides.

Compound	MIC (μg/mL)
**1**	64 ^a^
**2**	64 ^a^
**3**	4 ^c^
**4**	16 ^b^
Methicillin	2 ^d^
Chloramphenicol	4 ^c^

Data are presented as the mean of three parallel experiments. Different superscript letters (i.e., a–d) indicate significant differences among the compounds at *p* < 0.05.

## Data Availability

Data and figures generated or used in this study appear in the submitted article.

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
