# Peer review of "Bioassay-Guided Isolation of New Flavonoid Glycosides from *Platanus* × *acerifolia* Leaves and Their *Staphylococcus aureus* Inhibitory Effects"

_molecules, 2022, doi:10.3390/molecules27175357_

Round 1

Reviewer 1 Report

Abstract should be structured, consist background, methodology, results and conclusion.

Methodology should clearly briefed so that the researchers can guide.

Author Response

Point 1: Abstract should be structured, consist background, methodology, results and conclusion.

Response: Yes, thanks a lot! Abstract section has been improved accordingly.

Point 2: Methodology should clearly briefed so that the researchers can guide.

Response: Yes, the methodology has been rewritten in more details so that other researchers can follow up.

Reviewer 2 Report

Abstract: This section needs some development.

Figure 2 : the structures are not clear. '' ...NMR correlations of compounds 1 and 2 '' why you made the bold for 1 and 2 in this legend of Figure 2

Table 1: Please correct the bold in the title ''...1...''.

Table 2: Please correct the bold in the title ''...2...''.

Table 3: where the distribution (mean±SD).

In vitro evaluation (antibacterial activity): The chemical solution of DMSO is a product that has an antibacterial effect unless there is a low concentration which is not mentioned in this section. In addition, this section should be expanded.

Author Response

Point 1: Abstract: This section needs some development.

Response: Thanks! Abstract section has been improved.

Point 2: Figure 2: the structures are not clear. '' ...NMR correlations of compounds 1 and 2 '' why you made the bold for 1 and 2 in this legend of Figure 2?

Response: Thanks for pointing out this issue! Some of the non-essential HMBC correlations have been removed, along with the COSY correlations being recolored to black. However, we would like to retain the bold format for the numbers in the legend of Figure 2 as it is convention in chemistry related publications.

Points 3 and 4: Table 1: Please correct the bold in the title ''...1...''. Table 2: Please correct the bold in the title ''...2...''.

Response: Please refer to the reason listed in response to Point 2. We would like to keep them in bold.

Point 5: Table 3: where the distribution (mean ± SD).

Response: In this study, we don’t have the mean ± SD. Indeed, minimum inhibitory concentration (MIC) was evaluated on the basis of Clinical Laboratory Standards Institute (CLSI) guidelines by the conventional two-fold microbroth gradient dilution assay. In this study, the resulting working concentrations in 96-well plates were 512, 256, 128, 64, 32, 16, 8, 4, 2, 1 μg/mL for mixtures and 64, 32, 16, 8, 4, 2, 1, 0.5. 0.25, 0.125 μg/mL for single compounds, respectively. Basically, MICs are determined as the lowest concentrations of the drugs that completely inhibit the growth of bacteria, which means the three wells of one sample at the same concentration without any bacterial growth are used to obtain the final MIC. Therefore, the presentation of mean ± SD is not suitable for MIC values.

Point 6: In vitro evaluation (antibacterial activity): The chemical solution of DMSO is a product that has an antibacterial effect unless there is a low concentration which is not mentioned in this section. In addition, this section should be expanded.

Response: Yes, exactly. For the in vitro antibacterial assays with the concentrations of DMSO higher than 1%, controls are always added to evaluate the effects introduced by DMSO, which is every bit as what we have done in the present study. To make things clear, we also described the procedures in the Materials and Methods section in our initial submission. As to the expansion of the bioactivity study, the Mechanism of Action (MOA) is currently in progress in our laboratory, which has been stated in the Conclusions section and would pave the way for the further study.

Reviewer 3 Report

This is an exploratory descriptive study that discovered potentially useful antibiotics required to treat S. aureus infections. The technical quality and analytical methodology are good. The paper is well-written with concise conclusions and clarity.

A limitation for a peer reviewed research paper is the lack of statistical evidence that the work is based on randomization and replication to eliminate variation that may occur from one collected sample compared to another. There are no statistical analyses to confirm significance. How many samples from different trees and /or leaves were collected. How many sub samples in analytical  data and  what variance was recorded? Was the work repeated?

Was there any variation among collected samples. Eventho the data (Table 3) show large differences, the authors are encouraged to provided one or more statistical parameters to show that differences are significantly different from the controls and among treatments.

L62: suggest "of" vs toward's subfractions.

L59: How many trees were sampled, how many leaves collected, what time of year and or growth phase were recorded?

L165: use ground instead of grinded. How many samples were prepared from how many leaves, was this repeated?

L197: medium twice, delete for

L215: Use; still lacking, vs lack of phytochemical...

Author Response

Point 1: A limitation for a peer reviewed research paper is the lack of statistical evidence that the work is based on randomization and replication to eliminate variation that may occur from one collected sample compared to another. There are no statistical analyses to confirm significance. How many samples from different trees and /or leaves were collected. How many sub samples in analytical data and what variance was recorded? Was the work repeated?

Response: Thanks a lot, this is a very impressive question. The main purpose of the present study aims at seeking bioactive secondary metabolites from plants, which is in the very much early stage of the natural product research related any drug R&D. We have only investigated one batch of leaves detailed in the manuscript. Statistical analyses regarding samples from different locations, season, etc. absolutely need to be included during the R&D. However, this move is considered to set aside until the near-late stage of the whole drug R&D process, or at the very least put in place after the mechanism of action (MOA) of those bioactive compounds being fully elaborated. In order to reflect the thoughts behind this reviewer comment in the revised manuscript, we have added a sentence to the Conclusion section. In addition, based on the MOA data having had in hand, the statistical analyses for sure will go deeper

Point 2: Was there any variation among collected samples. Eventho the data (Table 3) show large differences, the authors are encouraged to provided one or more statistical parameters to show that differences are significantly different from the controls and among treatments.

Response: Yes, the variation does exist. We have performed the variation analysis for the MIC results among the collected compounds. The superscript letters (i.e., a-d) are used to indicate the differences as shown in Table 3. Meanwhile, a paragraph (i.e., section 3.4) was added to describe the statistical analysis.

Point 3: L62: suggest "of" vs "towards" subfractions.

Response: Thanks! Revised accordingly.

Point 4: L59: How many trees were sampled, how many leaves collected, what time of year and or growth phase were recorded?

Response: In general, it is a very interesting question but we feel sorry that natural product chemists worldwide always do not pay much attention to how many leaves collected from how many trees when they do a phytochemical investigation. For this piece of work, all the details of the samples can be found in the updated section “3.2 Plant material” in the manuscript.

Point 5: L165: use ground instead of grinded. How many samples were prepared from how many leaves, was this repeated?

Response: Thanks! “grinded” has been replaced by “ground”. Of the sample information, please refer to the Point 4 above and the section “3.2 Plant material” in the updated manuscript.

Point 6: L197: medium twice, delete for

Response: Yes, this minor error has been corrected.

Point 7: L215: Use; still lacking, vs lack of phytochemical...

Response: Thanks. Revised accordingly.

Round 2

Reviewer 3 Report

The statistics added help considerably to gain confidence in the results. It will be interesting to see if the collection experimental design iin the next phase of the work impacts the results.